# Numerical simulation of magma-rock interaction at Krafla volcano using OpenFOAM software and a simplified thermal model

Muriel Gerbault<sup>1</sup>, Oleg Melnik<sup>2</sup>, and Anastassia Borisova<sup>1</sup>

<sup>1</sup>GET (IRD, CNRS, CNES, UPS, OMP), 14 av. E. Belin, Toulouse 31400, France

<sup>2</sup>Earth Science Department, University of Oxford, UK

**Correspondence:** Oleg Melnik (oleg.melnik@earth.ox.ac.uk)

**Abstract.** We present a 2D numerical modelling study aimed at exploring magma-rock interaction following the emplacement of a magmatic sill into cold shallow crust. An interface-tracking solver was developed, based on the open-source OpenFOAM package that enables simulation of heat and momentum transfer between magmas of different compositions, with contrasting densities, thermal properties, temperatures, crystal contents, and strain-rate dependent viscosities. Two scenarios are considered to reconstruct sharp temperature gradients and explain the presence of fresh rhyolitic fragments excavated from approximately 2 km depth during IDDP-1 drilling at Krafla caldera in 2009; partial melting of felsic crust triggered by either (1) a 300 m thick rhyolite intrusion or (2) a 100 m thick basalt sill. We also assume two possible magma emplacement periods, during the Krafla Fires (1975–1984,  $\sim$ 35 years before drilling) and during the Myvatn Fires (1724–1729,  $\sim$ 300 years before drilling). In scenario (1), vigorously convective molten rhyolite produces a temperature jump (400°C) over approximately 25 meters (~16°C/m) 35 years after emplacement. After 300 years, the thickness of these molten rocks reaches approximately 70 m, however, the thermal gradient becomes too small (less than 5 °C/m) to explain the IDDP-1 observations. In scenario (2), because of large density contrasts between the injected basaltic magma and molten rhyolite, two separate convective layers are formed. The thickness of molten rocks reaches about 40 m after 35 years, and the propagating melting front produces a sharp temperature gradient in the undisturbed rocks, greater than 26°C/m. These results together with previous petrological studies lead us to conclude that this second scenario of a basaltic intrusion provides a more robust explanation for the extreme geothermal gradient encountered in 2009 than the first scenario. By comparing with a simplified 1D thermal model and performing parametric tests, we argue that both 2D and 1D numerical approaches help constraining better magmatic convection at such extremely high Rayleigh and Prandtl numbers.

#### 1 Introduction

In several volcanic areas around the world, magma bodies were accidentally penetrated by drilling with the aim of discovering deep supercritical hydrothermal resources (e.g. Teplow et al., 2008). One of the first wells was drilled in the Krafla geothermal field in 2008–2009 by the Iceland Deep Drilling Project (IDDP-1). The well was designed to reach supercritical conditions at 4500 m depth, but at ~2000 m depth, drilling became difficult due to a sharp increase in temperature. Finally, drilling stopped at 2096 m depth, and cuttings of fresh rhyolitic glass indicated the presence of a magma body at the bottom.

**Figure 1.** Conceptual setting of Krafla, Iceland: right, a 3D view displays IDDP-1 and KJ-39 drilling sites approaching a magma body at ca. 2 km depth (red), over geophysical anomalies located at depths greater than 3 km (blue-to-purple). Left: this magma body is composed of partially molten host felsic rock above a (convecting) magmatic sill intrusion. We will not consider the overlying hydrothermal system here.

In the Menengai caldera in Kenya, geothermal drilling began in 2011, and since then multiple wells have penetrated syenitic magma 2 km below the caldera floor (Mbia et al., 2014). Not only were these magma bodies surprisingly shallow, but none of these magmas have erupted in recent times. At both the Krafla and Menengai sites, there was an abrupt transition from solid rock to molten magma with an extreme temperature gradient, as predicted by Carrigan (1984).

Krafla, one of the five volcanic systems in Iceland's North Volcanic Zone, recently recorded two major eruptive events: the Myvatn Fires in 1724–1729, and the Krafla Fires in 1975–1984 (e.g. Hollingsworth et al., 2012). While the 2009 IDDP-1 drilling showed magma at 2 km depth, seismic wave attenuation indicated a magma storage zone at rather 3 to 7 km depth (Schuler et al., 2015; Einarsson, 1978; Kim et al., 2020), coinciding with the geodetic-modelled depth to an inflating/deflating body during the Krafla Fires (Tryggvason, 1984), Fig. 1 (blue-to-purple domain). The spatial extent and dimensions of this magma body at 2 km depth remain poorly constrained, as well as its composition, whether basaltic or felsic. According to Eichelberger (2020), the temperature increased from 500 to 900°C within only 25 meters during the IDDP-1 drilling. Such a high temperature gradient (>16°C/m) requires active melting of the crust. The presence of a sharp temperature gradient as witnessed by the IDDP-1 drilling indicates convective heat transfer, because otherwise, conductive heat transfer would have rapidly smoothed it over time. In addition, while the area has been well monitored since the 1940s (with first triangulation and levelling techniques), no significant signal has indicated any kind of subsequent intrusion during the period in between the Krafla Fires and the IDDP-1 drilling in 2009, the only recorded displacement being a stable regional subsidence at a rate of approximately 1 cm/year, observed between 1993 and 2005 and attributed to both thermal and stress relaxation processes due

to the Krafla fires (Drouin et al., 2017). This thermal anomaly can therefore reasonably be attributed to the Krafla fires or to older intrusive events.

According to drilling logs (Mortensen et al., 2014), the crust above the magma body in Krafla has a felsic composition. The magma uplifted by the drilling fluid also has a high silica content and ranges from crystal-free rhyolites to partially molten felsites, also called granophyres (Borisova et al., 2023). The debate about the nature of the intruding magma body is long-standing (e.g. Zierenberg et al., 2013; Rooyakkers et al., 2021), regarding whether not only basaltic but also rhyolitic magma intrusions caused the production of the felsic magma drilled in both the KJ-39 and IDDP-1 wells between 2008 and 2009 (Mortensen et al., 2010). Bimodal compositions of the erupted products were noticed since Grönvold and Mäkipää (1978), with lavas north of the caldera being more primitive than those inside the caldera during the first three eruptions (1975–1977). Presently, petrological studies indicate that a single reservoir cannot explain this bimodality (e.g. Rooyakkers et al., 2021). The recent study of Rooyakkers et al. (2024) invokes the necessity of short basalt–rhyolite mixing timescales (e.g., hours or days) and the ascent of both primitive and evolved basaltic magmas, driven by episodes of plate-boundary rifting during the Krafla Fires.

55

Available experimental studies of Icelandic rhyolite crystallization and felsite partial melting demonstrated the efficiency of partial melting of felsite crust (Masotta et al., 2018); Masotta et al. (2018) suggested that the IDDP-1 rhyolite magma was formed by high-degree partial melting of a quartzo-feldspathic rock ("granophyre") at shallow depth. On the other hand, a numerical model of assimilation (Simakin and Bindeman, 2022) established the kinetics of convective dissolution due to compositional convection (where the reaction is controlled by convective transport); a hot silicic magma that intrudes another cold, felsic crystalline rock (a "granophyre") at shallower crustal level may induce in it sufficient degrees of partial melting so as to trigger compositional convection there (Simakin and Bindeman, 2022). Simakin and Bindeman (2022)'s model only covers the first 100 days of evolution of the magmatic system after rhyolite injection, and these authors advocate that heat from a basalt intrusion would be necessary to keep the system hot enough for another 35 years. Based on analyses of the IDDP-1 zircons, Borisova et al. (2023) considered the viability of this scenario but suggested instead that the intrusion of basalt into shallow felsic crust followed by its melting, was a more plausible scenario to produce the magma extracted during the IDDP-1 drilling. These authors proposed that a magmatic sill of basaltic composition in its superheated state (at temperature above the magma liquidus temperature) intruded into and interacted directly with the predominantly felsic crust during the Krafla Fires. With the help of a one-dimensional (1D) thermochemical model, Borisova et al. (2023) explored the parametric range for a basaltic sill to produce the observed high temperature gradients  $\sim$ 35 years after the intrusion. As a follow-up numerical study, the present contribution aims to compare the two propositions of a felsic or a basalt intrusion triggering convective heat transfer in overlying crust, with the support of two-dimensional (2D) and 1D thermo-mechanical models. We present below the physical and numerical assumptions, then the resulting thermal evolution and the models limitations. Our results help improve our understanding of such magmatic dynamics and provide constraints for further drilling projects (e.g. Eichelberger et al., 2020).

Figure 2. Sketch of geometry and mean temperature profiles, modified after (Huppert and Sparks, 1988):  $T_o, T_\infty, T_m$  are initial hot, top, and melt temperatures respectively, D the intrusion thickness and a the molten host rock layer thickness. a) Case of rhyolite intrusion similar to the composition of the host rock (scenario 1), b) Case of basaltic intrusion into rhyolitic host rock (scenario 2).

# 75 2 Physical formulation and numerical setup

The problem of basaltic magma-felsic rock interaction was studied both experimentally and analytically by several authors (Huppert and Sparks, 1988; Carrigan, 1984). A review of different aspects of this problem can be found in a special collection "Magma-Rock and Magma-Mush Interactions as Fundamental Processes of Magmatic Differentiation" published in Frontiers (Borisova et al., 2021). Vigorous convection in a magma body intruded into cold host rock can induce advective heat transfer and melting of the overlying rocks. In these studies, magma intrusion is assumed instantaneous and its temperature is assumed to remain constant in space due to intense convective mixing. A thermal boundary layer (TBL) forms due to rapid cooling of the magma at the interface with the cold host rocks (Fig 2, between depths D and D + a).

If the hot intruded magma is rhyolitic and identical to the host rocks, a single convective cell develops and its upper boundary progressively propagates upwards (the melting front). This layer of molten host rock maintains a uniform temperature, while heat transfer remains conductive above it. If the melting front propagation rate is faster than the conductive timescale, a sharp temperature gradient develops across the molten/unmolten boundary (Fig 2, left at depth D+a). After some time, the whole system cools down and convection shuts down, leading to the retraction of the melting front and the return to a conductive heat profile.

In the case of a basaltic magma intrusion, the density contrast between the intruded magma and the molten host rock remains 0 high, preventing intense mixing, and a two-layer convection structure forms as shown in Fig. 2.

#### 2.1 System of equations

In order to model the physical setting presented above, we consider a numerical approach that solves the Navier-Stokes equations for incompressible fluids, with temperature- and crystal content-dependent viscosities, able to account for both compositional and thermal convection and for the release of latent heat of crystallization. We neglect volatile dissolution during basalt crystallization because the measured water content of the magma and the crust are below the saturation limit. See section 4.1 for further discussion.

The physical problem is described with the following equations of conservation of momentum, temperature and continuity:

$$\rho \frac{\partial \mathbf{U}}{\partial t} + \rho \mathbf{U} \cdot \nabla \mathbf{U} = -\nabla P + \rho \mathbf{g} + \nabla \cdot [\mu (\nabla \mathbf{U} + (\nabla \mathbf{U})^T)], \tag{1}$$

100 
$$\rho C_p \left[ \frac{\partial T}{\partial t} + \mathbf{U} \cdot \nabla T \right] = \nabla \cdot (k \nabla T),$$
 (2)

$$\nabla \cdot \mathbf{U} = 0. \tag{3}$$

U stands for the velocity field, P for pressure, T for temperature, g for gravity,  $\rho$  and  $\mu$  are density and dynamic viscosity. Thermal conductivity k and heat capacity  $C_p$  vary with temperature. Further details are provided in section 2.2.

We adapted the *multiMeltInterFoamv2* solver (Louis-Napoleon et al., 2020, 2022; Louis-Napoleon et al., 2024) based on the VOF (Volume of Fluid) method implemented in the open-source platform OpenFOAM; this method was shown to track well the evolution of distinct immiscible fluid phases and, therefore, solves an additional conservation equation for material interfaces  $C_i$  (dimensionless, here material phases are i = 1 for rhyolite and i = 2 for basalt):

$$\frac{\partial C_i}{\partial t} + \mathbf{U} \cdot \nabla C_i = -\nabla \cdot (\mathbf{U}_r C_r),\tag{4}$$

The  $-\nabla \cdot (\mathbf{U}_r C_r)$  term aims at reducing the effects of numerical smearing of phase interfaces, with  $C_r = C_1 \cdot (1 - C_1)$  and  $U_r$  a 'compression velocity', evaluated as a volume flux based on the maximum velocity magnitude in the interface region (Berberović et al., 2009). See Louis-Napoleon et al. (2022) for details and method validation.

#### 2.2 Mechanical and thermal properties

We assume that density does not depend on pressure but depends on temperature (the Boussinesq approximation) and on melt fraction M, with reference densities for the solidus and liquidus states of each phase (see Table 1 for values):

$$\rho = \rho_{ref} \times [1 - \alpha (T - T_{ref})], \text{ with } \rho_{ref} = \sum_{i=1}^{2} C_i \times [\rho_i^S (1 - M) + \rho_i^L M]. \tag{5}$$

Melt fraction M is parametrized for each phase i as:

$$M = (1 + a_i + b_i T_r + c_i T_r^2 + d_i T_r^3)^{-1}.$$

$$T_r = T/T_{ref_i}, T_{ref} = 1000^{\circ} C,$$
(6)

with parameters obtained by fitting simulation results of the crystallization of basaltic and rhyolitic magmas with typical Krafla composition (Borisova et al., 2023), using the MELTS software (Gualda et al., 2012).

The thermal conductivity k and heat capacity  $C_p$  are prescribed as:

$$C_p = C_{p_0} + L_* \frac{dM}{dT}, k = \frac{k_0}{1 + k_T T}, \tag{7}$$

with  $L_*$  the latent heat of crystallization,  $k_0$  and  $k_T$  constants given in Table 1.

Dynamic viscosity  $\mu = \mu_m(T)\eta(\phi,\epsilon)$  is a product of melt viscosity  $\mu_m$  given by Giordano et al. (2008) and a relative viscosity  $\eta$  due to the presence of crystals, that depends on the melt fraction M and the strain rate  $\epsilon$  according to Costa et al. (2009):

$$\log \mu_m = A_i + \frac{E_i}{T - T_i},$$

$$\eta(\phi, \epsilon) = \frac{1 + \phi^{\delta}}{[1 - F(\varphi, \epsilon, \gamma)]^{B\phi_*}}$$

$$F = (1 - \xi) \cdot \text{erf} \left[ \frac{\sqrt{\pi}}{2 \cdot (1 - \xi)} \varphi \cdot (1 + \varphi^{\gamma}) \right] \quad \text{with } \varphi = \frac{\phi}{\phi_*}, \phi = 1 - M.$$
(8)

Constants  $A_i, E_i, T_i$  differ between rhyolite and basalt phases (i), parameters  $\delta, \phi_*, B, \xi, \gamma$  are strain-rate dependent. Further details are provided in Appendix A.

## 2.3 Dimensionless parameters and boundary layer thicknesses

140

Convective heat transfer modelling is widely used in industrial, environmental and Earth Sciences applications over a broad range of spatial and temporal scales. Patterns of convection strongly depend on the geometry, boundary conditions, heat sources and material properties, and they are usually characterized by Rayleigh (Ra) and Prandtl (Pr) dimensionless numbers. A good review of existing experimental and numerical results in a Pr-Ra representation and convection regimes can be found in Silano et al. (2010). Convection in magmatic chambers is characterized by high Pr and Ra values located in the upper right corner of the regime diagram Fig. 4 from Silano et al. (2010). The corresponding regime is characterized by irregular transient convection, with narrow plumes of hot and cold magma detaching from the upper and lower boundaries between the magma and the host rock. The presence of distinct magmas (basalt and rhyolite) requires fine mesh resolution and expensive computational resources, hence a correct scalability of the problem is needed. Below we estimate the required mesh size in order to resolve the TBL thickness based on the values of parameters listed in Table 1.

The Rayleigh number Ra represents the ratio of the buoyancy force to dissipation forces. If  $\Delta \rho$  and  $\Delta T$  are the density and temperature contrasts across a layer of thickness H, g is the gravity acceleration,  $\kappa$  is thermal diffusivity and  $\mu$  the dynamic viscosity, Ra can be expressed in two ways:

$$Ra_T = \frac{\rho \alpha g \Delta T H^3}{\kappa \mu}, Ra_\rho = \frac{g \Delta \rho H^3}{\kappa \mu} \tag{9}$$

Prandtl's number characterizes the ratio between the thicknesses of the viscous and the thermal boundary layers,  $Pr = \mu C_p/\kappa$ . Both scenarios of a hot rhyolite or a hot basalt intrusion are characterized by extremely large Ra and Pr numbers. The intensification of heat transfer with respect to conduction is characterized by the Nusselt number,  $Nu = hH/\kappa$ , where h is a

| symbol                  | physical quantity                      | units           | range/initial values                                                  |
|-------------------------|----------------------------------------|-----------------|-----------------------------------------------------------------------|
| T                       | Temperature                            | °C              | $T_{\infty} = 400^{\circ}, T_o^r = 980^{\circ}, T_o^b = 1200^{\circ}$ |
| $T_{ref}$               | Reference Temperature (eq.6)           | °C              | $T_{ref} = 1000$                                                      |
| U                       | Velocity                               | m/s             | -                                                                     |
| x, y                    | Horizontal & Vertical coordinates      | m               | -                                                                     |
| H                       | Domain size $H \times H$               | m               | $H^b = 225, H^r = 750$ m                                              |
| D                       | Intrusion thickness D                  | m               | $D^b = 100, D^r = 300$ m                                              |
| $\alpha$                | Thermal expansion (eq.5)               | $K^{-1}$        | $3.10^{-5}$                                                           |
| $	ilde{ ho},  ho_{ref}$ | Local and reference Densities (eq. 5)  | $kg/m^3$        | 2300 - 3000                                                           |
| $\rho_i^S$              | crystal density of $i$ (eq.5)          | $kg/m^3$        | $\rho_r^S = 2700, \rho_b^S = 3000$                                    |
| $ ho_i^L$               | melt density of $i$ (eq.5)             | $kg/m^3$        | $\rho_r^L = 2300, \rho_b^L = 2800$                                    |
| $k_i$                   | Thermal conductivity range $i$         | W/m/K           | 1-2                                                                   |
| $k_o, k_T$              | Thermal conductivity constants (eqs.7) | $W/m/K, K^{-1}$ | 3,0.002                                                               |
| $L_*^i$                 | Latent heat of crystallization (eqs.7) | J/kg            | $3.5 \cdot 10^5$                                                      |
| $Cp_i, Cp_o$            | Heat capacity in i (eqs.7)             | J/kg/K          | $Cp_o^r = 1200, Cp_o^b = 1000$                                        |
| $\mu_i = \nu_i.\rho_i$  | Dynamic viscosity                      | Pa.s            | -                                                                     |
| $ u_i$                  | kinematic viscosity                    | $m^2/s$         | $0.5-10^{12}$ , see test cases                                        |

**Table 1.** Variables and Parameters of the models. Indices <sup>r</sup> for rhyolite, <sup>b</sup> for basalt and *i* for both recursively.

heat transfer coefficient. The intensity of momentum transfer is characterized by the Reynolds number,  $Re = \rho U H/\mu$ . At high Re convection is turbulent, while at low Re the flow pattern is laminar but can be highly transient.

Grossmann and Lohse (2000, 2001) theoretically related Ra-Pr and Nu-Re numbers over a wide range of values, and discussed implications on the Boundary Layer (BL) thickness, hence the characteristic length scales  $\lambda$  that impose a minimum mesh resolution to the models. They determine  $\lambda_U = L/(4\sqrt{Re})$  and  $\lambda_T = L/(2Nu)$  as kinematic and thermal length scales, with Re and Nu the Reynolds and Nusselt numbers respectively, and L = H/2 the characteristic thickness of the magmatic intrusion. For  $Pr > 10^6$  such as in the present problem, Nu becomes independent of the value of Pr and can be approximated by fitting the data presented on Fig. 2(a) from Grossmann and Lohse (2001) by:

$$\log_{10}(Nu) = 0.0087\log_{10}(Ra)^2 + 0.1350\log_{10}(Ra) - 0.2943. \tag{10}$$

The Reynolds number that corresponds to high Pr and Ra numbers (region  $III_{\infty}$  in Fig. 1 from Grossmann and Lohse (2001)) can be approximated as  $Re = 0.015Ra^{2/3}Pr^{-1}$ . Table 2 shows characteristic values for typical dimensionless parameters and estimates of the thickness of inertial and thermal boundary layers.

Therefore, we use characteristic density difference  $\Delta \rho$  and temperature difference  $\Delta T$  that fall between the 50% crystal state and the molten state, for each rhyolite and basalt phases (Marsh, 1981). Analysis of Table 2 reveals that convection is mostly driven by the magma crystallization/melting process, associated with large density variations. Re numbers are small due to high magma viscosity, meaning that the inertia effects remain insignificant and that the flow is mainly controlled by the com-

| Parameter                              | Symbol        | Rhyolite            | Basalt             |
|----------------------------------------|---------------|---------------------|--------------------|
| Temperature contrast ( ${}^{\circ}C$ ) | $\Delta T$    | 200                 | 200                |
| Density contrast $(kg/m^3)$            | $\Delta \rho$ | 200                 | 200                |
| Thermal Rayleigh number                | $Ra_T$        | $9 \cdot 10^{9}$    | $2 \cdot 10^{10}$  |
| Compositional Rayleigh number          | $Ra_{ ho}$    | $1 \cdot 10^{11}$   | $2.5\cdot 10^{11}$ |
| Prandtl number                         | Pr            | $4.9 \cdot 10^{8}$  | $6.6 \cdot 10^{8}$ |
| Nusselt number                         | Nu            | 178                 | 235                |
| Reynolds number                        | Re            | $6.8 \cdot 10^{-4}$ | $9 \cdot 10^{-4}$  |
| Thermal boundary layer                 | $\lambda_T$   | 0.42 m              | 0.11 m             |
| Kinematic boundary layer               | $\lambda_U$   | 1436 m              | 418 m              |

**Table 2.** Estimated dimensionless parameters and minimal thicknesses of the thermal and kinematic boundary layers in rhyolite and in basalt, assuming that relevant variations in temperature and density stand above 50% crystals Marsh (1981). Other parameters taken from Table 1.

petition between the viscous resistance and buoyancy. In turn, the large Pr numbers impose the strongest constraints on the mesh resolution, to resolve the thickness of the thermal boundary layer.

According to Stevens et al. (2013), "The best way to confirm that the used numerical resolution is sufficient is to obtain the same Nusselt number with different grid resolutions as there is namely always some uncertainty in estimates of the required grid resolution". We will show below that the simulated heat released from the magma becomes mesh-size independent when the mesh size becomes comparable to the thickness of the TBL ( $\lambda_T$  and Nu have a linear inverse relationship).

#### 170 2.4 Numerical Setup and parameter ranges

The model setup is a 2D square domain of dimension  $H \times H$  made of rhyolite crust (r subscript parameters) at a uniform temperature of 400°C, in the middle of which a hot magma intrusion of thickness D and Temperature  $T_o$  is emplaced (see setup Fig. 3). In scenario 1, the rhyolite intrusion has a thickness D=300 m at temperature  $T_o=980$ °C, and the domain's size is H=750 m. In scenario 2, the intrusion is composed of basalt of thickness D=100 m at  $T_o=1200$ °C and the domain's size is H=225 m. The top boundary has a free-slip condition while the bottom boundary has a no-slip velocity condition, and temperatures are maintained there, fixed at 400°C. Velocities and temperatures at the lateral borders are set periodic.

### 3 Model tests and results

165

The 2D models show how a rhyolite or a basalt intrusion progressively melts the cooler crust above, producing a sharp temperature gradient at the melting front, with the setup provided above (Fig. 3). Complementary tests illustrate the influence of mesh resolutions, viscosities, and domain sizes on the evolution of this melting front; they are listed in Table 3 and are described in greater detail in Appendix B. Here we present the results around a main best resolution case for each scenario, and synthesize the importance of these numerical artifacts. A comparison with a 1D model complements this analysis.

| Case               | size(m)/cells | $\min(\nu_b - \nu_r)$ | $\max(\nu_b - \nu_r)$ | Time | Figures      |
|--------------------|---------------|-----------------------|-----------------------|------|--------------|
| R4                 | 750/182       | 150                   | 1e11                  | 250y | 6a, B1,      |
| R2                 | 750/375       | 150                   | 1e11                  | 500y | 6a, B1       |
| R1                 | 750/750       | 150                   | 1e11                  | 500y | 6a, B1       |
| R0.5               | 750/1500      | 150                   | 1e11                  | 250y | 4, 5, 6a, B1 |
| B1.0_v2_m1-10      | 225/225       | 2 - 150               | 1e10 - 1e11           | 50y  | 6b           |
| B0.5_v0.5_m5-50    | 300/600       | 0.5- 150              | 5e10 - 5e11           | 30y  | B5           |
| B0.5_v2_m1-10      | 300/600       | 2 - 150               | 1e10 - 1e11           | 30y  | 6b,B1,B5     |
| B0.5_v2_m0.5-0.5   | 225/450       | 2 - 150               | 5e9 - 5e9             | 40y  | В6           |
| B.45_v2_m5-50      | 225/500       | 2 - 150               | 5e10 - 5e11           | 50y  | 6b,B1,B2,B5  |
| B.45_v10_m1-10     | 225/500       | 10 - 150              | 5e10 - 5e11           | 28y  | 6b,B5        |
| B.45_v100_m1-10    | 225/500       | 100 - 150             | 1e10 - 1e11           | 32y  | В5           |
| B.45_v1000_m1-10   | 225/500       | 1000- 1000            | 1e10 - 1e11           | 65y  | В5           |
| B.45_v2_m1-10_x10  | 225/500       | 20-1500               | 1e11 - 1e12           | 40y  | 6b,B1,B5     |
| B.45_v2_m1-10_x100 | 225/500       | 200-15000             | 1e12 - 1e13           | 60y  | B1,B5        |
| B.25_v2_m1-10      | 225/900       | 2 - 150               | 1e10 - 1e11           | 25y  | 6b,7,8,B1,B5 |
| B.25_v10_m1-10     | 225/900       | 10 - 150              | 1e10 - 1e11           | 25y  | 6b,B1,B5     |
| B.25_v10_m1-100    | 225/900       | 10 - 150              | 1e10 - 1e12           | 23y  | B1,B5        |
| B.225_v2_m1-10     | 225/1000      | 10 - 150              | 1e10 - 1e11           | 12y  | 6b,B1        |
| B.225_v10_m1-10    | 225/1000      | 10 - 150              | 1e10 - 1e11           | 50y  | 6b,B1        |
| B.125_v10_m1-10    | 225/1800      | 10 - 150              | 1e10 - 1e12           | 20y  | B3,B4        |
| BT20.5_v1_m1-20    | 250/500       | 1 - 150               | 1e10 - 2e11           | 30y  | C1           |
| BT50.4_v2_m5-100   | 200/500       | 2 - 150               | 1e11 - 1e12           | 30y  | C2           |

**Table 3.** Model cases, testing the influence of mesh resolution and viscosity ranges. The model name logic is R(meshsize) for Rhyolite cases and B(meshsize)\_v(vismin)\_m(vismax) for basalt intrusion cases, with "vismin" the minimum basalt viscosity and "vismax" the maximum basalt-maximum rhyolite viscosities factor of  $10^{10} m^2/s$ . The last two "BT" cases test the basaltic sill thickness (20 m and 50 m). "Time" is the maximum time until which the model could run.

**Figure 3.** Numerical setup for a) a basalt or b) a rhyolite intrusion. Boundary conditions in temperature (T) and velocity (U) displayed in grey. c) melt fraction of magmas depends on temperature according to eqs.(8) for basalt and rhyolite (red and blue curves respectively).

# 3.1 2D Rhyolitic sill intrusion (scenario 1)

Our first scenario considers the injection of a rhyolitic sill. Figure 4 displays snapshots in time of the temperature, melt fraction and velocity, and zooms of other variables near the melting front. Figure 5 displays the temporal evolution of the melting front, velocity and averaged temperature profiles. Since the hot intrusion domain is initially set at 980°C, it convects. Over time, this domain progressively "shifts" upwards, with the rhyolite melting front propagating upwards and the base of the intrusion cooling down and adopting a conductive regime. This model illustrates the following key features of heat transfer:

- The melting front propagates upwards reaching an additional thickness of about 25 m after 35 years, and then a thickness of 70 m, 200 years following intrusion emplacement.
- After 35 years, a temperature gradient of 450°C occurs over a depth range of about 30 m (between depths 370 and 400 m, Fig.4, blue curve), providing a geothermal gradient of 15°C/m. After 300 years, this temperature jump occurs over a depth range of more than 150 m (the geothermal gradient attains 3°C/m, purple curve).

The melting front boundary develops sharp density, conductivity and heat capacity gradients. After 35 years these contrasts reach ca. 10% for the conductivity, 16% for the density, and 50% for the heat capacity (Figure 4, top). After 300 years the domain of molten rhyolite has shifted up by 70 m, but the system is cooling down with much smoother temperature gradient, density and conductivity gradients (Figure 4, bottom). Heat capacity still varies greatly due to the influence of latent heat, but over a thinner ~10 m thick layer. The molten zone starts to shrink down to ca. 70 m from about 300 years (Figure 5 and video in Supplementary material).

Comparison of the melting zone thickness (MZT) over time for different mesh resolutions is illustrated in Figure 6a. We see that for mesh sizes smaller than 1 m, the MZT evolves close to each other with differences reaching 16% (12 m) after ca. 50

**Figure 4.** Rhyolite intrusion 300 m thick (scenario 1, model R0.5): In the centre, there are snapshots in time of the temperature (left column), melt fraction (middle) and velocity field (right) over the entire domain size. Dashed lines indicate the initial location of the hot intrusion, and help see heat propagating upwards, together with the melting front and the velocity field. Figures in boxes on the top and bottom are close-ups near the upper melting front displaying temperature, density, heat capacity, velocity after 35 and 300 yr, respectively.

Figure 5. Rhyolite intrusion (scenario 1, model R0.5): Melt front thickness (top, thickness over the initial emplacement depth at y=350 m) and velocities (middle, in  $log_{10}$  scale) over time, and average temperature profiles with depth (bottom). Note that the melting front thickness starts to decrease from ca. 300 yr; the 400°C jump occurs over 25 m after 35 yrs, then over more than 150 m after 300 yrs after emplacement.

years, but decreasing to less than ca. 2% after 100 years. We propose that critical mesh size has been reached and that the heat transfer is modelled correctly.

During the IDDP-1 experiment, the temperature increased by more than 100°C over several meters: in comparison in this modelled scenario 1, the temperature increases by about 450°C over about 30 m (15°C/m), 35 years after emplacement of the intrusion (e.g., assuming emplacement during the 1984 Krafla fires). This is just about equal to the gradient that was observed. It becomes more difficult to attribute the observed thermal gradient to a rhyolite body intruded 300 years ago, since the model indicates that the sharp thermal gradient then spreads over a thickness of 150 m, corresponding to a temperature increase of 3°C/m (Figure 5, bottom). Hence the motivation to attempt a model case with a basaltic intrusion, cf. next section.

**Figure 6.** Mesh resolution tests for a) a rhyolite intrusion (scenario 1) and b) a basalt intrusion (scenario 2): melt zone thickness over time with respect to its initial top location. Numbers in legend refer to cell size and viscosities, see Table 3 for further details. The model cases in blue in (a) and in red in (b) are the ones displayed in Figs. 4-5, and Figs. 7-8, respectively.

## 210 3.2 2D Basaltic sill intrusion into felsic crust (scenario 2)

For scenario 2 we assume that a basaltic sill of 100 m at 1200°C is injected into a cold rhyolite crust. The results are displayed in Figures 7 and 8: in Figure 7 the thermal gradient between molten basalt and cold rhyolite (the dashed line represents its initial location), shows how the rhyolite layer progressively melts, with "independent" convection developing between the convecting basaltic layer and the conductive host rock above.

Figure 7 shows that a layer of partially molten granophyre (rhyolite) has already formed 5 years after injection, and it expands steadily during the following decades (see also video in Supplementary material). Within that layer, the density contrast reaches 500 kg/m³ and heat capacity is boosted by latent heat, exceeding in effect values of 4000 J/K. Velocities are slow at the beginning and then increase and vary with spikes related to the convection dynamics (Figure 8). From about 15 yrs onwards we see that the velocities start to reduce, indicating that the maximum rate of heat transfer has passed; the system cools down despite the melting zone thickness (MZT) still increases. At the basalt-rhyolite boundary, a plateau in temperatures follows the initial sharp thermal jump. This plateau corresponds to the convective layer of partially molten rhyolite. It tends to smooth out towards a conductive profile over time (cf. Figure 8):

After 5 years, the temperature across molten/unmolten rhyolite jumps from about 500 to 1000°C over a depth range of about 10 m (between depths ca. 160 to 170 m), which corresponds to a thermal gradient of 50°C/m,

**Figure 7.** 2D Model with a basalt intrusion (scenario 2, model B.25\_v2\_m1-10): center figures show snapshots in time of temperature (left), melt fraction (middle) and velocity field (right). The dashed lines represent the initial upper boundary of the intrusion. Top-bottom figures zoom at the melting front boundary: temperature, density, heat capacity and velocity, after 10, 20 and 25 yr.

**Figure 8.** Basalt intrusion (scenario 2, model B.25\_v2\_m1-10): Melt zone thickness (MZT) and velocities over time, and average temperature profiles with depth, up to 25 years.

- After 25 years (Fig.7c), the temperature increases from 500 to 950°C over ca. 15 m (between depths ca. 175 to 190 m), which corresponds to a thermal gradient of about 30°C/m.
- The melting front thickness (the plateau width) reaches ca. 30 m after 25 years, and still displays an increasing trend.

This high resolution case crashed after 25 years. We conducted a number of additional tests with moderate success (see also the discussion section). Lower resolution tests were faster computationally and could run for longer times, but lost accuracy. Figure 6b displays the MZT evolution for our most successful attempts, and confirms the first-order influence of the mesh cell sizes on melting front propagation rates. A tendency can be identified: the difference in the melting zone thickness reaches 30% for a cell size change from 1 m to 0.5 m (MZT 20 m vs. 27 m); it reduces to 15% for cell sizes change from 0.5 to 0.25 m (MZT 27 m vs. 32 m), after 25 yr; it reduces to less than 3% (barely visible) when cell sizes are further reduced, but those

test cases didn't reach 25 yrs. Furthermore, this difference in MZT is maintained over time in the 3 tests that attained 50 years (22% and 14% in between these different cell sizes, respectively).

Additional tests of the influence of the modelled domain size and of the minimum and maximum viscosities are displayed in Appendix B, and show that the melting front thickness (MZT) is affected in the following ways:

- Tests with a wider domain (500 × 600 m) show no significant difference in MZT with the reference case (225 × 225 m), indicating that the chosen modelled height and width do not influence the results (Appendix Fig. B5a).
- Higher maximum viscosities for both rhyolite and basalt domains have a minor effect on the results, within a range of values of 10<sup>10</sup> 10<sup>12</sup> m²/s. Greater values do not impact the system's dynamics given the overall time-scale of the processes at play. Lower rhyolite viscosity destabilizes the system gravitationally but does not appear physically realistic, given a) the known petrophysical properties of the host granophyre rock in that area and b) the significantly smoother geothermal gradient that is produced then (cf. discussion).
- We also tested the minimum viscosity (for basalt), since it affects the numerical time-step (reducing it reduces the time-step). Our comparisons in Appendix Fig. B5 show that the melt front propagation rate and the system's velocities vary by less than 10% when the minimum kinematic viscosity ranges from 0.5 to 10 m²/s. In contrast, the melting zone thickness and the velocities are reduced by about 20% when the minimum viscosity is multiplied by a factor 100.

Given the results described above, we achieved an ultimate test at a resolution of 0.125 m approaching the critical-mesh size of 0.11 m, and a minimum viscosity of 10 m<sup>2</sup>/s (about 10 times greater than the real minimum viscosity of basalt). This run achieved a time duration of 20 years and displays MZT values well superimposed on the previous test cases with lower minimum viscosity and coarser mesh resolution (cf. Figures 6b, B3, B4). This model case ran for 3 months on 36 Intel® Skylake 6140 - 2.3 Ghz cores, and then locked into an increasingly smaller time-steps regime. Nevertheless this similarity in the resulting MZT evolution gives us confidence that the values obtained at a mesh resolution of 0.25 m but achieving longer modelled times would be satisfactory, cf. Figure 6b and discussion section).

Since the numerical mesh resolution is found to have such a strong impact on the modelled results, this 2D case is further compared with a 1D modelling study in the next section.

#### 3.3 Comparison between 2D simulations and a simplified 1D model

260

Modifying the approach of Huppert and Sparks (1988), Borisova et al. (2023) developed a 1D thermochemical model of heat transfer from a convecting basaltic intrusion into host felsic rocks, reproducing its melting and the production of hot rhyolite magma. This 1D thermochemical model is based on a non-linear heat conduction equation which accounts for the release of latent heat of crystallization and for convection (via effective thermal conductivity) in a three-layered system with contrasting compositions. The middle layer contains initially hot basaltic magma that releases heat and melts the surrounding felsic rocks. We use the same phase diagrams and rheological models for basalt and rhyolite as in the full 2D simulations. The model is

265 described by the following system of equations for each layer i = 1, 3:

$$\rho_i C p_i \frac{\partial T}{\partial t} = \frac{\partial}{\partial x} k_i \frac{\partial T}{\partial x}; C p_i = C p_i^0 + \frac{dX_i}{dT} L_*^i;$$

$$N u_i = \frac{Q_{conv}}{Q_{cond}} = G(Ra_i); k_i = k_m(T) \cdot N u_i$$
(11)

Here the dependence of the Nu number on the Ra number  $(G(Ra_i))$  is described by eq. (10).  $k_m(T)$  is the molecular (or local) conductivity.

**Figure 9.** Comparison between 2D (dots) and 1D (solid line) averaged temperatures at 5, 10, 20 and 30 years, for the basaltic intrusion (scenario 2).

Each layer is divided into several zones representing thermal boundary layers and the core of the flow, where intense convective mixing occurs. According to 2D simulations very little melting occurs at the bottom contact between basaltic magma and the crust. Instead, the solidification front propagates in basaltic magma leading to a decrease in the effective width of the convective layer. We use the analytical solution of Stefan's problem to calculate the position of this front as:

$$x_s = \sqrt{\frac{\kappa \cdot t}{Ste}}, Ste = \frac{Cp^b \Delta T}{L_*^b}$$
 (12)

where  $\Delta T = 300^{\circ}$ K gives the best fit to the front position (see Fig. 9).

280

In the middle of the convective layers the effective thermal conductivity is parametrized based on the Nu-Ra relationship approximated by fitting the data from Grossmann and Lohse (2001) for extremely large Pr numbers, cf. eq. 10 (section 2).

Fig. 10 shows that after the intrusion of basaltic magma the effective thermal conductivity can exceed the "molecular" thermal conductivity by up to 3 orders of magnitude. This leads to a uniform temperature distribution within the molten magma. As the basaltic layer cools down the value of effective conductivity decreases progressively. At the upper contact between basaltic magma and molten rhyolite, two TBLs are formed, one in each magmas. Their thicknesses are calculated using the assumption that the  $Ra_{\delta}$  number defined by the thickness of the TBL  $\delta$  is equal to the critical value,  $Ra_* = 1708$  for the onset of convection in a horizontal layer. A similar TBL originates at the boundary between molten rhyolite and the host rock which is defined by the position of the maximum value of the temperature gradient  $\frac{\partial T}{\partial x}$ .

Figure 10. Effective thermal conductivity at 5, 10, 20 and 30 years.

Overall, there is good agreement between the predictions of the 1D and 2D models in terms of the propagation of the melting front and temperature gradient in unmolten rocks. The 1D model shows temperature gradients within convective regions, meaning that the estimated value of the effective thermal conductivity is not large enough. That might be a consequence of strain-rate dependent viscosity used in the 2D modelling, while the 1D modelling uses an averaged value. The high Nu numbers obtained in the 1D simulations are close to the values shown in Table 2, suggesting that both 1D and 2D models capture the convective heat transfer correctly.

## 290 4 Discussion

300

In this work, for the two modelled scenarios of the cooling of a basaltic or rhyolite intrusion, we have obtained a propagating melting front through initially cold rhyolite within ca. 30 years or more, that displayed sharper temperature gradients in case of a basaltic intrusion than in case of a rhyolitic intrusion. Several aspects of this modelling are discussed below, separated into numerical aspects first and then aspects related to petrological observations from Krafla.

#### 295 4.1 Numerical aspects and potential complementary factors at play

We have tested the influence of the numerical mesh size, and concluded that in case of a rhyolitic intrusion (Scenario 1) we achieved a mesh size (50 cm) that approaches sufficiently the minimal thickness of the thermal boundary layer (40 cm) since the melting zone thickness (MZT) tends to become mesh-size independent within 10% error. In that case, the MZT achieves 25 m, 35 years after emplacement, and achieves 70 m 300 years after emplacement. In case of the basaltic intrusion (Scenario 2) our highest resolution test (cell sizes of 12.5 cm) also approaches the theoretical thickness of the boundary layer (11 cm, cf. Table 2) and we show that the MZT evolution with time displays nearly superimposed curves when the mesh size attains 0.25 m and greater. Hence we find that in that scenario as well, numerical convergence is achieved. With a mesh cell size of 0.25 m, we are able to predict a MZT of about 40 m and a temperature gradient of 44°C/m, 35 years after emplacement (Appendix B).

We could have also tested a variety of intrusion thicknesses and more complicated shapes than just a flat layer, a three-dimensional body instead of an infinitely long layer as implied by the 2D assumption; this would also directly affect the melting front thickness, but such thickness and shape are not well constrained by geophysical data. A rhyolite intrusion of thickness greater than the 300 m assumed in Scenario 1 would manage to reproduce the observed geothermal gradient, but it cannot be too thick or else it would have been detected by geophysical methods. For a basaltic intrusion in turn, we chose an a priori thickness of 100 m, based on previous estimates by e.g., Borisova et al. (2023). We display in Appendix C, two extra cases with a 20 m thick and 50 m thick basalt intrusion. The 20 m thick intrusion appears too thin to generate convection in the overlying rhyolite (Figure C1) and the thermal profile remains conductive in that case. The 50 m thick intrusion manages to generate convection within 20 m of the overlying rhyolite (Figure C2), with a temperature gradient similar to observations. Hence a basaltic sill 50 m thick can also be a realistic scenario.

In order to approach the required mesh size in our models, many cases actually crashed for Scenario 2, because either the run entered into a locked regime with decreasing time-steps tending towards 0, or the time-steps would increase too fast that the VOF approach diverged and the run stopped. To achieve reasonable modelled time duration and computational times, we had to increase the minimum viscosity of the basaltic material and therefore we tested the influence of modifying viscosity ranges. We discussed and showed in Appendix B, that below a kinematic viscosity of 10 m²/s the results display similar behaviour and nearly indistinguishable melting front propagation rates. Besides, augmenting the maximum viscosity does not have a significant impact, given the time-scales considered here (of course longer modelled time periods would require to consider greater maximum viscosities). On the other hand, a model in which the host rock maximum viscosity was empirically reduced by a factor 5 (Fig. B6) showed that the partially molten host rock destabilizes gravitationally within about 20 years, producing diapirism and a significantly smoother temperature gradient than the one observed during the IDDP-1 drilling. De facto, this extreme case shows that our choice of maximum viscosity of the host rock in the other models was appropriate.

Finally, one may ask whether the active hydrothermal system above Krafla's magma reservoir or volatile exsolution from the magma influences the evolution of the melting processes modelled here. The water content has been evaluated to stand below 1.9 wt%  $H_2O$  according to analyses by (Pope et al., 2013) of Krafla fresh lavas, and displays values between 0.18 and 1.22 wt%  $H_2O$  in the granophyre rocks (Liu et al., 2005). Furthermore the water content of IDDP-1 samples is also found to be below the water saturation level at 2 km depths ( $\sim$  54 MPa - a pressure at which rhyolite melt at water saturation should contain more than 2 wt%, Borisova et al., 2024). Some volatile exsolution can occur only at late stages of basalt crystallization due to residual melt enrichment in volatile elements.

Furthermore, the crustal rocks at Krafla were found strongly heterogeneous with respect to <sup>18</sup>O<sub>VSMOW</sub> (Vienna Standard Mean Ocean Water, Borisova et al., 2024). This indicates that meteoric fluids interacted with these crustal rocks before they melted and re-crystallized, and that hydrothermal circulations did not impact directly crust melting process. This information comforts us in our assumptions of parameters (viscosity, melting laws) corresponding to an overall anhydrous context. Nevertheless, the thermo-mechanical-chemical coupling between the hydrothermal and magmatic domains is known to be complex, and at present day, most advanced modelling studies of magma and hydrothermal fluids interactions still rely on strong approx-

imations and empirical thresholds for crystal content and/or permeability, because these are difficult to precisely constrain in the field (e.g. Gruzdeva et al., 2024).

The geothermal gradient obtained in our models with low resolution (0.45 m) reaches ca. 27°C/m after 35 years of basalt-rhyolite interaction, and still exceed 16°/m after 50 years (fig. B4a). The resolution case at 0.25 m produces 44°C/m. In order to explain potentially even greater geothermal gradients, and let aside the option of a thicker initial intrusion, one would need to take into account more localized, reactive fluid flow processes, such as reactive porosity waves (Wong and Keller, 2023), brittle behavior (Witcher et al., 2025) or specific drilling-induced localized flow (Wadsworth et al., 2024).

Simulations of high Ra, high Pr number thermochemical problems remains challenging. Not all our simulations reached the final time. The time-steps diminishes during the first ca. 10 years of the modelled run-time, and in successful cases it manages to rise back up again once the convective layers are stably growing. Several cases end up with extremely small time-step, others in drastic oscillation in the time-step that finally leads to unphysical results. Bounding the time-step upper limit promotes better stability. Higher mesh resolution tends to produce more stable runs despite longer computational duration, indicating that small time and lenght-scale perturbations are properly resolved. At later stages when basalt becomes viscous and the viscosity contrast between the magmas becomes small the mixing intensity increases and, of course, this process is not captured by the VOF method. A full complementary "THMC" (thermo-hydro-mechanical-chemical) modelling study will be needed to tackle the magma dynamics below Krafla, hopefully in the near future.

### 4.2 Complementary arguments from analyses of Krafla's rock samples

From the two principal scenarios that were simulated here, the numerical results indicate that the thin basaltic intrusion scenario matches the temperature jump observed during the IDDP-1 drilling experiment, slightly better than the thick rhyolitic intrusion scenario, across the propagating melting front that separates convective and conductive heat transfer modes.

With a basaltic magma intrusion the density contrast between the intruded magma and the molten host rock remains high, prevents intense mixing, and a two-layer convection structure forms as shown in Figure 7. Below we describe the petrological studies that further strengthen our preference for this scenario rather than for the one assuming a rhyolitic magma intrusion.

The main question about basaltic versus rhyolitic intrusions interacting with the Icelandic crust is still debated, although our previous study (Borisova et al., 2023) and the current investigation of the Krafla zero-age material from IDDP-1 and KJ-39 sites samples (Borisova et al., 2024) suggest a direct participation of basaltic rather than rhyolitic magma in their production. Our thermodynamic modelling of equilibrium crystallization of the Krafla granophyre performed using rhyolite-MELTS (Borisova et al., 2024) at 900–1150°C predicts that clinopyroxene, orthopyroxene, sodic plagioclase An<sub>10-20</sub>, K-feldspar, quartz and magnetite spinel, rutile are stable at 0.1 GPa and at the QFM (quartz-fayalite-magnetite) buffer. These mineral phases (except rutile) are observed both in the Krafla felsite samples and in the IDDP-1 glassy samples, suggesting that they attained physical-chemical conditions close to thermodynamic equilibrium due to high degrees of partial melting. These conditions are also in agreement with experimental data on partial melting of the Krafla granophyre (Masotta et al., 2018).

From a petrological view point, the interaction of basaltic magma with felsite crust (or "granophyre") is recorded by the presence of accessory mineral phases, such as baddeleyite, together with zircon and chevkinite and by significant glass hetero-

geneity with respect to Si, Fe, Ca and  $^{18}O_{VSMOW}$  in the IDDP-1 samples (Borisova et al., 2024). Oxygen isotope data measured by Hampton et al. (2021) in the IDDP-1 samples provide low  $\delta^{18}O_{VSMOW}$  (+3‰), further indicating that they cannot have formed due to segregation of pure partial melt alone, but rather formed as a result of the interaction of basalt and rhyolite.

The magmatic crystals in the Víti felsite (the granophyre rock that erupted from the Víti crater during the Myvatn Fires; the Víti crater being itself located near the IDDP-1 drilling site) appear almost mantle-like (e.g., pyroxene IC-82; +4.75‰), while other magmatic crystals have much lower <sup>18</sup>O<sub>VSMOW</sub> values that rather reflect assimilation of hydrothermally altered material (e.g. plagioclase IC-83; -5.35‰). This strong isotopic heterogeneity can be directly explained by multi-stage rock-basalt-fluid interaction in the granophyre. Different batches of silicic melts were likely produced during the partial melting of the host granophyre rocks, which was originally strongly heterogeneous due to hydrothermal fluid circulations and alteration. In other words, the partial melting of quartz-feldspar-rich granophyre rock and its hybridization with basalt explains the composition of this hybrid magma. All this information corroborates our numerical models in favor of the scenario of a basaltic intrusion during the 1975–1984 Krafla Fires, to produce the rhyolite magma encountered during the IDDP-1 drilling.

# 5 Conclusions

In this work, we have modelled in two dimensions the cooling of a basaltic and a rhyolite intrusion over decades, and we have obtained values of the propagating melting front through the initially cold rhyolite that lies above it. We have shown that our numerical models satisfactorily approach the required critical numerical resolution for the convecting, partially molten boundary layer. The thickness of this propagating melting zone was obtained in both two cases, one with an initially 300 m thick rhyolitic intrusion (Scenario 1) and one with an initially 100 m thick basaltic intrusion (Scenario 2). In scenario 1, the models produce molten rhyolitic zone thicknesses of about 25 meters for a rhyolitic intrusion that occurred 35 years after the Krafla Fires, reaching 70 meters if this intrusion occurred 300 years after the Myvatn Fires. However, after ca. 300 years the thermal gradient has significantly smoothed out, making us rule out this option. In scenario 2 with a basaltic intrusion, we obtain molten rhyolitic thicknesses of about 40 meters, 35 years after the Krafla Fires, reaching 50 meters 50 years after the Krafla Fires. These values of the molten rhyolitic thickness depend directly on the dimensions of the initial intrusion and other unknown physical properties, hence, they may vary with respect to the range given above. After 35 years, the models display sharper temperature gradients in the case of a basaltic intrusion (44°C/m) than in the case of a rhyolitic intrusion (15°C/m). While a basaltic intrusion easily reproduces the geothermal gradient observed during the 2009 IDDP-1 drilling, a rhyolitic intrusion might also be plausible if it was more than 350 m thick; but then it would be detected by geophysical methods (which has not yet). Hence while a rhyolite intrusion may fit observations within a rather narrow range of parameters, a basaltic intrusion covers a wider range of parameters. These results corroborate other petrological arguments indicating that the felsic crust drilled in 2009 at the IDDP-1 site was molten in contact with a basaltic intrusion from below. Therefore, we prefer Scenario 2 over Scenario 1.

These models offer useful information for future drilling exploration projects approaching partially molten bodies and brittle-ductile transition depths, but shall be further refined with next generation models accounting for more complex physical behavior taking into account multiphase flow and chemical mixing.

*Code availability.* The OpenFOAM solver, reference case input files and videos are available in Supplementary Material and on the Gitlab repository (upon signing in): https://gitlab.com/AurelieLN/MultiMeltInterFoam. See the associated user guide for further details.

Data availability. All data are presented in the main text and appendices, as well as on the Gitlab repository: https://gitlab.com/AurelieLN/MultiMeltInterFoam.

Video supplement. Videos are available as Supplementary Material.

Author contributions. Conceptualization: OM, MG, AB. Formal analysis: MG,OM. Funding acquisition: AB. Investigation: MG, OM, AB. Methodology: OM, MG. Software: MG,OM. Resources: AB. Writing of the original draft: OM, MG, AB. Writing review and editing: OM, MG, AB.

Competing interests. The authors declare that they have no conflict of interest.

- Acknowledgements. This work was supported by the program PAUSE from Collège de France and Wolfson Royal Society visiting fellowship (RSWVF-R1-231013) for O.M. This article is funded by the European Union (ERC, PLANETAFELSIC, project 101141259) to A.B. Views and opinions expressed are however those of the author(s) only and do not necessarily reflect those of the European Union or the European Research Council. Neither the European Union nor the granting authority can be held responsible for them. Numerical models were carried out on the CNRS regional supercomputer center CALMIP (https://www.calmip.univ-toulouse.fr/) under project number p24041. AnneMarie
   Cousin (GET) drew Figure 1. Discussions with Simon Rooyakkers (GNS) and Thomas Bonometti (IMFT) greatly helped mature the study.
  - Reviews by Catherine Annen and Alain Burgisser greatly helped improving this work.

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

# Appendix A: Magma rheology

Here we provide more detail on the magma rheology, following Giordano et al. (2008); Costa et al. (2009). The viscosity of the melt phase is calculated based on the Vogel–Fulcher–Tammann (VFT) equation (Giordano et al., 2008):

$$\log \mu_m = A_i + \frac{E_i}{T - T_i},\tag{A1}$$

where for basaltic magma  $A_b$ = -9.6,  $E_b$  = 1.33e4,  $T_b$ = 307.8 K, and for rhyolitic magma  $A_r$ = -8.15,  $E_r$  = 2.40e4,  $T_r$ = -431 K. These parameters are calculated for the typical melt compositions reported in Borisova et al. (2023) and in Borisova et al. 2025 (submitted), and are provided in Table A below.

|                      | Krafla Fire basalt | Krafla (Viti) granophyre |
|----------------------|--------------------|--------------------------|
| SiO <sub>2</sub> wt% | 49.98              | 74.71                    |
| $TiO_2$              | 2.16               | 0.47                     |
| $Al_2O_3$            | 13.21              | 11.95                    |
| $Fe_2O_3$ tot        | 15.95              | 3.66                     |
| MnO                  | 0.25               | 0.08                     |
| MgO                  | 5.15               | 0.35                     |
| CaO                  | 9.35               | 1.84                     |
| $Na_2O$              | 2.56               | 4.06                     |
| $K_2O$               | 0.40               | 2.65                     |
| $P_2O_5$             | 0.21               | D.L.                     |
| $H_2O$               | 1.22               | 0.13                     |
| $CO_2$               | 0.11               | 0.08                     |
| $\delta^{18}$ O, %o  | ~5.6               | -1.8  to  4.7            |

Costa et al. (2009) proposed a model for the relative viscosity of the magma that depends on the second invariant of the strain-rate tensor  $\epsilon$ , and on the crystal content  $\phi = 1 - M$ , where M is the melt fraction:

$$\begin{split} &\eta(\phi,\epsilon) = \frac{1+\phi^{\delta}}{[1-F(\varphi,\epsilon,\gamma)]^{B\phi_{*}}} \\ &F = (1-\xi) \cdot \operatorname{erf}\left[\frac{\sqrt{\pi}}{2 \cdot (1-\xi)} \varphi \cdot (1+\varphi^{\gamma})\right] \quad \text{with } \varphi = \frac{\phi}{\phi_{*}}, \phi = 1-M, \\ &\phi_{*} = 0.066499 \tanh(0.913424 \log(\epsilon) + 3.850623) + 0.591806, \\ &\delta = -6.301095 \tanh(0.818496 \log(\epsilon) + 2.86) + 7.462405, \\ &\alpha = 1-\xi = -0.000378 \tanh(1.148101 \log(\epsilon) + 3.92) + 0.999572, \\ &\gamma = 3.987815 \tanh(0.8908 \log(\epsilon) + 3.24) + 5.099645. \end{split}$$

In the models, we can set as a multiplicator constant to scale all values (set to 1, 10 or 100 according to test cases), but also set minimum and maximum viscosity bounds; see the values assigned to each model case in Table 3.

## Appendix B: Numerical tests for mesh resolution and viscosity ranges

#### B1 Mesh resolution tests

We detail here the influence of mesh resolution on the evolution of the rhyolite melting zone thickness (MZT) over time (Fig. 6). We see that for the rhyolitic intrusion (Fig. 6.a), the MZT curves reach similar heights when the cell size becomes smaller than 1 m from 200 yrs onward, confirming that the maximum MZT likely to be attained is 70 m. However, after 35 years, one should rather rely on the highest resolution test (resolution 0.5 m) with a value of the MZT reaching ca. 25 meters thick, whereas the 1 m resolution test reaches only ca. 16 m (30% difference).

Figs. B1 display the velocity magnitudes for all tested cases. For the rhyolite intrusion, we note that mean velocities remain quite similar independently of the mesh resolution. The maximum velocities increase with mesh resolution but reach similar values for the 2 highest resolution cases  $(5.10^{-5} \text{ and } 3.10^{-4} \text{ m/s}, \text{ eg. less than } 10 \text{ m/yr}, \text{ for the mean and maximum velocities respectively), corroborating mesh convergence.}$ 

**Figure B1.** Mean and maximum velocities for model cases (ordinates in exponential scale of m/s). a) Rhyolite intrusion, for 4 different mesh resolutions. b) Basalt intrusion, for different mesh resolutions and different extrema viscosities. Only cases that could exceed 15 years are displayed. See Table 3 for details.

For the basalt intrusion, mean and maximum velocities displayed Fig. B1b decrease with increasing minimum viscosity. They remain little sensitive to the mesh size once it is smaller than 0.45 m and viscosity is smaller than  $10 m^2/s$ , standing in the range of  $3.10^{-4}$  and  $2.10^{-3}$  m/s, corresponding to about 10 and 60 m/yr, for the mean and maximum velocities respectively.

We display below for comparison with the case of Figure 7 at resolution 0.25 m, an additional model case with a smaller cell size of 0.45 m (B2and B4a) that reached 50 years while the previous one reached only 25 years. We also display the results for the highest resolution case (0.125 m, minimum viscosity  $10 \, m^2/s$ ) in Figures B3 and B4b, but that only achieved 20 yrs (and ran for 3 months on 36 cores of the regional cluster).

**Figure B2.** Basalt intrusion scenario (2) with greater mesh cell size 0.45 m (model B.45\_v2\_m5-50) than the case displayed in Figure 7 with cell size 0.25 m, for comparison: behavior is similar. Similar legend except that zoom snapshots are after 5 years and 30 years here.

**Figure B3.** Basalt intrusion scenario (2) with highest resolution of 0.125 m (model B.125\_v10\_m1-10), and which only reached 20 years. Snapshots display temperature, viscosity and velocity field, and in zoom panels the conductivity and heat capacity.

Figure B4. Basalt intrusion scenario (2), from to to bottom, plots over time of the rhyolite Melting Zone Thickness, mean and maximum velocities and horizontal averages of the temperature with depth, for **a**) the moderate resolution of 0.45 m and minimum viscosity of 2  $m^2$ /s that reached 50 years (case B.45\_v2\_m5-50), and **b**) the highest resolution of 0.125 m and minimum viscosity of 10  $m^2$ /s (case B.125\_v10\_m1-10), that reached 20 years.

According to Fig.6.b, we see that a resolution of 25 cm shows an increase of the MZT by about 5 meters with respect to cases with resolution 45 cm, indicating a 15% difference. But in models at a resolution greater than 25 cm, the MZT is hardly distinguishable and at most 2% higher than in cases at resolution 25 cm, showing that we are approaching critical mesh size.

Given the curve shape of the temporal evolution of the MZT for the rhyolite case, where its slope decreases and flattens from ca. 250 years onward (Figure 6a), and considering that in the basalt test cases, mean velocities decrease from 20 years by nearly a factor 10, and continue decreasing, we can estimate that the MZT in the basaltic case will less than double its thickness over the next decades as the system cools down to a conductive state. In fact, in the three cases that achieved 50 years of modelled time at three different mesh resolutions, the MZT evolves in equal proportions than at shorter times and, for the best resolution (0.25 m), we are confident that the resulting prediction of 40 m after 35 years is robust (Fig. 6b).

Regarding the thermal gradient across the rhyolite melting front, in scenario 2 with a basalt intrusion, the 0.45 m resolution case displays values of 26.8°C/m, while the 0.225 m resolution case displays 44°C/m after 35 years (at depths around 180 m).

# **B2** Testing viscosity ranges

560

565

For the basaltic intrusion we also tested the influence of the minimum and maximum viscosities, and the values chosen in this study result from various preliminary tests. The lower limit was not varied since it is already a realistic value obtained from the melting laws given in Appendix A. The upper limit in turn was tested: the higher this value the greater the viscosity contrast but, a large value is useless given the time-scale of the modelled process (decades, much shorter than a higher viscosity relaxation time). Choosing the viscosity of the cold bedrock must still be high enough or else it can destabilize gravitationally (Fig. B6).

Figure B5. Influence of kinematic viscosity on the evolution of the molten rhyolite thickness (MZT) for the basaltic intrusion (scenario 2). a) Tests of the minimum viscosity of basalt from 0.5 to  $2 m^2/s$  and maximums  $(1-50.10^{10} m^2/s)$ , for a coarse mesh, showing similar results within that range. b) Tests of minimum viscosities from 2 to  $10^3 m^2/s$  and maximum viscosities from  $10^{10}$  to  $10^{13} m^2/s$ : they display similar behavior when minimum viscosities are 2 to  $10 m^2/s$  (at same mesh resolution), but "slower" MZT growth when they are greater.

The resulting melting zone thickness from our tests are plotted in Figures B5:

- Modifying the maximum kinematic viscosities of the rhyolite and basalt phases within a range of magnitudes does not significantly impact the results as long as they are greater than ca.  $10^{10}m^2/s$ . At smaller maximum values of viscosities, the entire rock mass destabilizes gravitationally and buoyant diapirs of crystallizing basalt develop into low viscosity partially melting rhyolite (cf. Fig. B6).
- Testing lower minimum viscosities for basalt was a challenge since some runs manage to go through and many more do not. The listed test cases are those that went through, indicating that a minimum basalt viscosity in the range of  $0.5 10m^2/s$  produces similar melting front behaviour.
- Greater overall kinematic viscosity values (multiplied by 100 or 1000 m<sup>2</sup>/s) significantly impact the system's velocities and dynamics, slowing down the melting front propagation rate.

Figure B6. Case with a 100 m thick basalt intrusion and a weak host rock (maximum kinematic viscosity reduced to  $5 \cdot 10^9 m^2/s$ , model B0.5\_v2\_m0.5-0.5): the latter destabilizes within ca. 15 years, producing diapirism of the partially molten host rock and smoothing out of the thermal gradient. This case may represent a mechanically weakened host rock due to hydrothermal fluid circulations, a hypothesis ruled out for the IDDP-1 context also based on petrological information (see Discussion section).

# 570 Appendix C: Thinner basalt intrusion tests

Several tests with a thinner basalt intrusion were conducted; we display here 1) a 20 m thick intrusion displaying conductive cooling after only 7 years, Fig. C1, 2) a 50 m thick intrusion developing a 15 m thick molten rhyolite layer still convecting after 30 years, Fig. C2.

Figure C1. Model with a 20 m thick basalt intrusion (BT20.5\_v1\_m1-20). a) 4 snapshots of the temperature, melt fraction and velocity fields. Colour bar ticks are for the last displayed time step (30yrs). b) Melting front thickness and velocity magnitudes over time, and temperature profiles over depth: they become conductive after  $\sim$ 7 years (velocity low and smooth temperature profile).

**Figure C2.** 50 m thick initial basalt intrusion (model BT50.4\_v2\_m5-100), for which secondary convection still occurs after 30 years, but only over ca. 20 meters. Legend same as previous figure. The 400 °C thermal jump after 30 years now occurs over ca. 25 m, corresponding to 16°C/m. Hence a 50 m thick intrusion may be just sufficient to explain the observed thermal gradient during the IDDP-1 drilling.