# Peer review of "Numerical simulation of magma-rock interaction at Krafla volcano using OpenFOAM software and a simplified thermal model"

_EGUsphere, 2025_

## Referee Comment (RC2)

This work presents numerical simulations of the temporal evolution of an instantaneously emplaced sill with the aim to reproduce temperature gradient observed in a natural drill hole.

I find this work to be a very nice and important follow-up study of the work of Borisova et al. (2023) that brings a very significant increment in physical accuracy. In this well-written manuscript that I enjoyed reviewing, the authors make a compelling case for this type of simulations. I have only two minor reservations. The first is that, although I understand that the authors would like very much like to use the results to discard one of the two explored scenarios (rhyolite sill vs. basalt sill), the first part of the discussion is too curt towards the rhyolitic sill scenario. I suggest ways to soften their assessment and honor the complexity of comparing temperature gradient across a notoriously complex interface. The second reservation concerns the limits of the model, which are very clearly stated, except for the reasons behind model crashes and the assumption of the absence of water saturation. Here also, I list below specific (and hopefully constructive) questions that should clarify these two points.

As all my other comments are directed towards clarifications and no additional runs are needed to complete this elegant and topical study, I recommend acceptance with minor revisions.

Detailed comments.

l. 14. This formulation about the choice of rhyolite vs. basalt sill is well-balanced and does not suffer from the same limitations as the beginning of the discussion (cf. comment on l. 285).

l. 16. I strongly suggest rephrasing this sentence, because the approach can only be qualified of "suitable" if it did not crash inexplicably after a few years.

l. 34-37. First, state the observation *According to Eichelberger (2020)…*, then mention the interpretations *Such a high temperature gradient …* and the *presence of this sharp gradient* […] *indicates convective transfer…*

l. 42. The names are confusing. IDDP-1 is the name of a drilling site (or project?) containing a well also numbered IDDP-1 and another well numbered KJ-39? Please clarify for the audience unfamiliar with the Krafla site.

l. 65 *setting -> setup*

l. 71 *constant due to -> constant in space due to*

l. 80 It seems that there is an unmentioned assumption of no fluid saturation. This is important because the rhyolite has 1.9 wt% $H_2O$ (see comment on l. 460). I suggest that you add that assumption here, with, if relevant, possible justification from previous works that neglecting fluids is a reasonable hypothesis.

Section 2.3 I appreciated the very nice preliminary assessment of the relevant scaling relationships of this system.

l. 163. The text mentions a no-slip velocity condition, whereas Fig. 3 mentions a free-slip condition. Please clarify.

l. 179-180. *temperature gradient -> temperature difference*. In fact, it would be much clearer to systematically compare temperature gradients explicitly, just as done in Scenario 2. It was hard to follow and gather information to see that Scenario 1 has a 15±0.25 °C/m gradient after 35 yrs, which is slightly lower (and mostly likely undistinguishable from, more on that later) than the observed gradient of ≥16 °C/m.

l. 186 I do not understand why the grid size selection is evaluated against the melt zone thickness, whereas the Section 2.3 states on l. 155 that the best measure is the Nusselt number. Please clarify why the finding of Stevens et al. (2013) is set aside here.

l. 190 Please give a quantified range of temperature gradient here for consistency (as stated currently, the range is ≥11–33 °C/m, which is compatible with the rhyolite sill gradient).

l.212 *°C/m/m -> °C/m*

l. 214 (and also l. 290) It is important to state the reason(s) behind the crash: vanishing time step, stalling of the residuals, temperature runaway, instability of Eq (4), … Currently, I can only infer that the crash probably does not occur because of too high a viscosity (see next comment).

l. 229 The maximum viscosity is said to have a minor effect on the results. Does it mean that it also does not affect crash time (see comment above)?

l. 265. *"although the 1D model shows temperature gradients within convective regions"*. This is a major drawback as the variable of interest is the temperature gradient. Just looking at Fig. 9, I guess that the 1D gradients across the melt zone thickness are basically meaningless because they are so far away from the 2D gradients. If my guess is correct, I suggest adding a few sentences about this issue as it shows clearly the added value of this 2D study.

l. 283. I did not understand the end of the sentence: by a comparable amount to what?

l. 285. I appreciate the discussion and evaluation of the effect of the basalt sill thickness. I wonder why the rhyolite sill thickness is not mentioned, and thus I suggest adding a few sentences about it. Borisova et al. (2023) tried with a sill thicker that the 300 m used here and found larger temperature gradients after the dreaded 35 yrs. I am not a fan of highlighting every parameter and asking to explore it further, but here the conclusion of the paper hinges on that single sill thickness. Presumably a sill of 350 m would bump the gradient from 15 to 16 °C/m, suddenly making Scenario 1 unquestionably valid?

l. 317 "best matches" is a simplistic assessment of the results. I suggest rephrasing that first § in terms of degrees of freedom and parameter ranges. As an example, here is what I have in mind: *An extrapolation of our results suggests that Scenario 1 likely fits the observed values within a very narrow range of parameters, whereas our results for Scenario 2 covers a wider range of parameters that yield gradient comparable to those observed. […] For all these reasons we prefer Scenario 2 over Scenario 1.*

l. 330-344. The petrology § are not linked to the framework of this study. Please ensure that it is the case. For instance, fractional crystallization is incompatible with the model assumptions, which needs to be mentioned. Another example is that hydrothermal fluids were needed to produce the partial rhyolitic melts, but 1) the model ignores hydrothermal fluids and 2) the whole § on l. 301-309 is dedicated to show that hydrothermal fluids played no *direct role in* […] *partial melting and the following reaction of the rock with* [putative] *basaltic magma*. Finally, the duration of 33 yrs is chosen here (and also in the Conclusions l. 354), whereas the whole work (starting l. 60) and all model results were evaluated at 35 yrs. This would be a detail if I were not tempted to wonder how much higher the 15 °C/m gradient of Scenario 1 is at 95% of the simulation time.

Appendices. I appreciated the clever selection parameter sweeps that gave me confidence in the numerical outputs.

l. 460. I was trying to find the answer to the question: how much, if any, total/dissolved water were assumed to be present in the rhyolite/basalt. Borisova et a., (2023) reports rhyolite composition with 1.9 wt% $H_2O$, and, unless I got lost, no basaltic composition. This is a confusing issue as this water-bearing rhyolite is the result of partial melting but the injected sill is a different (source) rhyolite in Scenario 1. To clarify this issue, could you simply add a Supplementary/Appendix Table with the initial MELTS composition for both Scenarios?

Alain Burgisser

---

## Author Response (AR1)

**Answers to Reviews**

**September 25, 2025**

We thank both reviewers for their careful reading and constructive remarks over our manuscript. Below are our detailed answers (in blue). We inserted and complemented our first replies from August 13th to reviewer 1 (RC1) and reviewer 2 (RC2). We find that now our work is even more robust and we hope that it is now suitable for publication.

**1 RC1: 'Comments by Catherine Annen, 25 Jul 2025**

**General comment**

The study presented in this manuscript seeks to account for the high temperature gradient revealed by IDDP-1 drilling in Krafla caldera. The study consists in a series of numerical models involving a basalt or rhyolite sill that is convecting and melts the felsic crust above. A best fit was found with observation with a 100m sill basalt although numerical issues were encountered depending on the chosen viscosities.

I like that the authors report both successful and failed models as negative results are also important to the advance of science. The results are interesting and coherent. The text is not always clear, but it is often more an issue with the style than with the science. I found the discussion particularly difficult to follow: it lacks a clear narrative and some sentences are non sequitur.

I have no major issues with the science as the model and its limitations are well explained.

Thank you for your comments.

**Specific comments**

The only point is that although I understand the use of the Myvatn Fires and Krafla fires to have some timescales for the model, I am not sure that there is any strong argument for the timing of the sill emplacement to coincide with an eruption. It might be the opposite with a sill emplacement being a failed attempt of the magma to reach the surface.

We think that an injection of a basaltic or rhyolitic sill after Krafla Fires is highly unlikely because the volcano is very well monitored, and such an event would definitely be recorded. We've failed to find any evidence of post Krafla Fires emplacement. The case of Myvatn Fires is less constrained, and there could have been undocumented failed eruptions before the Krafla fires. We precise now in the manuscript, lines 37-42, the study of Drouin et al. 2017 on Insar, GPS and leveling data spanning the period of 1995 to 2015: stable subsidence rates occur at ca. 1 cm/yr and are attributed to both the thermo-mechanical relaxation of the Krafla Fires event and the geothermal power-plant activities there.

**Detailed comments and corrections**

1.35-37: Couldn't the high temperature gradient also be explained by a very recent intrusion and heat that hadn't had time to diffuse?

That is related to the previous comment. Geophysical observation indicate that at least after the Krafla Fires, new emplacement is unlikely, given the lack of geophysical monitored evidence for that (eg. geodetic inflation/deflation).

l.49: can the crust be felsite? I thought felsite was for volcanic rock. Maybe "rock" would be better than crust here. Corrected. See also our clarifications further below. l.44 we first define "granophyre", and replaced l.59 "by a second felsite lens" by "by another felsic rock lens (forming another "granophyre")".

ll.44-45: I suggest removing "Until today". Can you be more specific as why a single reservoir cannot explain bimodality (just one or two sentence)? Does it mean that the rhyolite has to be melted crust but if bimodal reservoir was possible then the rhyolite could have come from the reservoir?

A recent injection of the rhyolite will not produce enough heat to melt the crust above the sill unless the thickness of a rhyolite sill is significant. Such sill should be detected by geophysical methods. It is not possible to sustain high temperature gradient without active melting of the crust. We inserted this comment in the discussion section now, see further answers below. We also rephrased lines 59, staying generic at this stage.

1.54 It is not clear what is the process advocated by Simakin and Bindeman. If it is precisely what has been described before, I suggest giving the reference and removing "corresponding to the process advocated by". Sentences have been rephrased there. In the discussion Simakin and Bindeman suggest that heat from basalt is necessary to keep the system hot for 35 years after Krafla fires.

1.95-100: What is the dimension of Ci?

Ci is the concentration of magmas in the VOF method, dimensionless. Inserted in the text.

In table 1, crystal densities are lower than melt densities, isn't it the other way round? Corrected, thank you.

l.130: I find confusing to have k for thermal diffusivity and  $\kappa$  for thermal conductivity as it is usually the other way round.

All switched.

1.163: there is a contradiction between text and figure regarding the bottom boundary condition: no slip vs free slip.

No slip, the figure was corrected.

Equation (5): The meaning of Ci is not clear to me and why the sum is 1 to 3, when above on l.96, i is 1 or 2. Is Ci in equation (8) the same? It seems to be a temperature there.

In eq. 5, i should be "2" indeed, that is corrected. In eq. 8,  $C_i$  is not the same as in eq. 5 indeed; and these are temperatures, we have renamed them as  $T_i$  there and in the appendix. Thank you.

1.197: add "a": injected into a cold rhyolite crust. Corrected.

1.206: "increasing" instead of increases. Why?

l.206: rephrase: a jump cannot form a plateau. Maybe something like: "A plateau in temperatures follows an initial sharp thermal jump at the basalt-rhyolite boundary. The plateau corresponds to the convective layer of partially molten rhyolite." Corrected (now l.212).

1.209 and 211: the term "evolves" is more suitable for a time variation than a spatial variation. A possible alternative phrasing could be: "After 5 years, the temperature across molten/unmolten rhyolites jumps from about 500 to 1000oC over a depth [..]". Corrected.

1.215 I suggest removing "Naturally". Corrected.

1.217 Instead of "tendency can be draw", "tendency can be identified". Corrected.

- l. 217: Needs rephrasing. If I understood correctly (not sure it is the case): "the difference in the melting zone thickness reaches 25% for a cell size change from 0.5 m to 1 m (MZT 20 m vs 27 m); it reduces to 15% for cell sizes change from 0.25 to 0.5 m, etc. Corrected; thank you (now 1.224).
- l. 223: I suggest replacing "Therefore one can expect it would actually reach [..] if we would reach the critical thickness" with "Therefore, we hypothesize that it would reach [..] if the critical mesh size could be reached". Rephrased, now 1.234-247.

1. 226: "that" instead of "how". Corrected.

1.260: the sentence is ambiguous. Is it 2 TBLs, one in each magma, or 4 TBLs, two in each magma? One in each magma in the upper contact of the basaltic sill with the cold rhyolite; corrected, 1.272.

1.263: "same": what is same referring to? is a "similar" TBL more clear? we replaced that word.

1.283: "this would also affect the melting front thickness by a comparable amount." I don't understand this sentence; comparable to what? We deleted this end of sentence and rephrased, l. 297. Actually any

geometrical variation would affect the melting front thickness by a factor proportional to the size of the variation itself with respect to the sill, but that remains to be demonstrated and it is not our purpose here to go in that detail.

l.284-285: I suggest rephrasing: "The intrusion's thickness controls the amount of melting of the overlying rhyolite but is not well constrained by geophysical data." Replaced, thank you (1.297).

1.290: The sentence sounds awkward, consider rephrasing. rephrased.

1.294: What does "greater modelling time mean" (longer, shorter)? Indeed, replaced by "shorter computational" times.

1.295-296: Sentence starting with "Beside" is not clear. We added "(of course longer modelled time periods would require to consider greater maximum viscosities)" (l. 316).

1.299-300: Awkward phrasing. A choice cannot be sufficient. Replaced by "appropriate".

1.302: "the water content of Krafla's fresh lavas to granophyres is low" I don't undersand. Do you mean fresh lavas and granophyres? or are fresh lavas granophyres? We reorganised this § and clarified indeed what were the analyses made on fresh lavas and those made on granophyres, see lines 317-321 now.

l.303: what does subscript VSMOW mean in relation with 18O. Please, clarify for non-experts. It means "Vienna Standard Mean Ocean Water", now precised in the text (line 322); it is an isotopic standard for water, characterising a reference water sample whose proportions of different isotopes of hydrogen and oxygen are accurately known and lacks salt or other impurities.

1.303-304. Sentence stating with "Thus" I do not understand what is meant. It sounds non-sequitur. This § was reorganised, now lines 316-327.

1.333-335: This sentence is difficult to follow and lack specificity: segregation of pure partial melt from where to where? What is meant by pure? Assimilation of what? Fractional crystallization of what?

The whole § was reorganised and these words were deleted., please see now lines 353-355.

1.336: What is the Víti felsite?we added there (now 1.356): (the granophyre rock erupted from the Víti crater during the Myvath Fires, the being located Víti crater near the IDDP-1 drilling site).

1.343: It is not clear how the last sentence of the section (starting with "In this context") links to the former statements (d18O heterogeneities). The sentence was rephrased and our reasoning reorganised in these paragraphs from lines 316 to 364, and we hope it is clearer now.

1.351-352: I am not convinced that the sill intrusion has to coincide with an eruption.

That is true. Let us recall our answer to a previous point, now inserted in the introduction, by citing Drouin et al., 2017, who showed that geodetic measurements since the Krafla Fires have not shown any post-Krafla Fires intrusive events (no inflation, just stable subsidence); that is why we (and others) assume that the last intrusive event occurred during the Krafla Fires (or before).

1.465: What do you mean by one can set a scaling factor. How does it link with eq. A2. Please, be more specific. We rephrased as "a scaling factor (set to 1, 10 or 100 according to test cases, as a multiplication constant in front of the viscosity law)".

We have tested two viscosity corrections to see the influence of this parameter. First a multiplication of the viscosity by the scaling factor, second - setting the upper and the lower bounds of the viscosity.

l.482: What is the curve shape? What curve? we rephrased as: "Following the curve shape of the MZT for the rhyolite case, in which the slope decreases and flattens from ca. 250 years onward (Figure XX) "...

1.498: "it can destabilize gravitationally" Is it really unrealistic? Do you mean unrealistic in this specific case or in general? It looks like stoping.

It depends on the timescale. If we model only 30-300 years of the system evolution, then the viscosity of host rocks can be set to an "unrealistical" low value  $\sim 10^{10}$  Pa s, because no significant viscous deformation occurs in the cold region of the domain anyway. If we were to study longer timescales, then the models would

require a higher viscosity threshold than the one chosen here. Actually, the  $\sim 400C$  surrounding bedrock is likely to have absolute viscosities exceeding  $\sim 10^{19}$  Pa s if we were to adopt conventional power-law creep rheology for felsic rock. But it is not useful to use that value here, given the short duration of the modelling time compared to viscous relaxation times. In turn, if the host rock viscosity is set to  $\sim 10^9$  Pa s, diapirs develop in the cold rocks as we show in Appendix, which is not realistic either given a) that it produces a much too smooth thermal gradient compared to the IDDP-1 observation, and b) that to justify such low viscosity, the samples should show more hydrous properties that what has actually been observed. This is discussed just below and in the discussion section in a clearer way now.

Hydrothermal circulations may reduce bulk rock effective viscosities (two phase viscous media tend to behave with a viscosity closer to the weakest phase), hence at some point we had thought that hydrothermal circulations may reduce the effective viscosity of the bedrock domain in our models. But in this specific Krafla context, petrology studies show that the rocks at depth become significantly anhydrous; this means that increasing depth and temperature tend to close the pore space, hence hydrothermal circulations at the near surface seem to be disconnected from the deeper, melting domain. Hence considering low viscosity bedrock appears irrelevant, e.g. irrealistic here.

Caption of figure 1: the geophysical anomaly is said to be purple but I see it blue. Corrected to "blue-purple".

Caption of figure 2: Have Tinf the same in caption and figure. Delete "hot"? a) Case of rhyolite -ADD/ with composition- similar to [..]. All corrected, thank you.

Figure 4: I am confused by the diagrams of velocity. The velocity seems to be non zero in areas that are solid. How is that?

Velocities of  $10^{-10}$  m/s result in total displacement of  $\sim 1$  m in 300 years. This is negligible in comparison with the domain size and displacements in the convective part of the domain.

Figure 5 and 8 top: why different colours for the symbols?. We changed them to black.

Figure 5 middle: We lose the value of mean velocity after 400 yrs. Would you consider changing the scale of Y axis? Done.

Caption figure B3: The last sentence sounds to incolloquial. the heading is expanded.

**2 RC2: Comments by Alain Burgisser, 20 Aug 2025**

This work presents numerical simulations of the temporal evolution of an instantaneously emplaced sill with the aim to reproduce temperature gradient observed in a natural drill hole. I find this work to be a very nice and important follow-up study of the work of Borisova et al. (2023) that brings a very significant increment in physical accuracy. In this well-written manuscript that I enjoyed reviewing, the authors make a compelling case for this type of simulations. I have only two minor reservations.

The first is that, although I understand that the authors would like very much like to use the results to discard one of the two explored scenarios (rhyolite sill vs. basalt sill), the first part of the discussion is too curt towards the rhyolitic sill scenario. I suggest ways to soften their assessment and honor the complexity of comparing temperature gradient across a notoriously complex interface.

We have softened our assessment in the discussion, conclusions and abstract, and aknowledge that the rhyolitic scenario cannot be totally ruled out. Thank you for helping us remain "objective".

The second reservation concerns the limits of the model, which are very clearly stated, except for the reasons behind model crashes and the assumption of the absence of water saturation. Here also, I list below specific (and hopefully constructive) questions that should clarify these two points.

We now provide details, below and in the text. Thank you for helping us clarify this point too.

As all my other comments are directed towards clarifications and no additional runs are needed to complete this elegant and topical study, I recommend acceptance with minor revisions.

Thank you very much Alain for this positive and motivating review. We detail below our answers.

**Detailed comments**

- l. 14. This formulation about the choice of rhyolite vs. basalt sill is well-balanced and does not suffer from the same limitations as the beginning of the discussion (cf. comment on l. 285).
- l. 16. I strongly suggest rephrasing this sentence, because the approach can only be qualified of "suitable" if it did not crash inexplicably after a few years. We replaced this term by "helps constraining better".
- 1. 34-37. First, state the observation According to Eichelberger (2020)..., then mention the interpretations Such a high temperature gradient ... and the presence of this sharp gradient [...] indicates convective transfer... These sentences order has been swapped.
- l. 42. The names are confusing. IDDP-1 is the name of a drilling site (or project?) containing a well also numbered IDDP-1 and another well numbered KJ-39? Please clarify for the audience unfamiliar with the Krafla site. Now the drilling sites are distinguished from the projects.
  - 1. 65 setting  $\rightarrow$  setup. Corrected.
  - 1. 71 constant due to  $\rightarrow$  constant in space due to. Corrected.
- l. 80 It seems that there is an unmentioned assumption of no fluid saturation. This is important because the rhyolite has 1.9 wt% H2O (see comment on l. 460). I suggest that you add that assumption here, with, if relevant, possible justification from previous works that neglecting fluids is a reasonable hypothesis. We have added a sentence on volatile saturation prior to equations 1.189.

Section 2.3 I appreciated the very nice preliminary assessment of the relevant scaling relationships of this system. Thank you.

- l. 163. The text mentions a no-slip velocity condition, whereas Fig. 3 mentions a free-slip condition. Please clarify. Corrected.
- l. 179-180. temperature gradient  $\rightarrow$  temperature difference. In fact, it would be much clearer to systematically compare temperature gradients explicitly, just as done in Scenario 2. It was hard to follow and gather information to see that Scenario 1 has a 15 $\pm$ 0.25 °C/m gradient after 35 yrs, which is slightly lower (and mostly likely undistinguishable from, more on that later) than the observed gradient of  $\geq$  16°C/m. Yes you are correct; we have now checked throughout the text that both gradients are expressed consistently and when similar one with another, we discuss them now more clearly. Cf. new abstract, discussion and conclusions sections, and further responses below.
- l. 186 I do not understand why the grid size selection is evaluated against the melt zone thickness, whereas the Section 2.3 states on l. 155 that the best measure is the Nusselt number. Please clarify why the finding of Stevens et al. (2013) is set aside here.

This statement by Stevens et al. is not set aside actually, since  $\lambda_T$  and Nu have a linear inverse relationship:  $\lambda_T = L/2.Nu$ . Studying the evolution of one directly corresponds to studying the other. We have detailed this now line 164, thank you for helping us clarify this.

Furthermore, we find that determining the melting zone thickness is a key outcome of the paper - how much molten rhyolite is produced. Nusselt number gives only first order estimates because the parametrization of Stevens et al. (2013) was done for a slightly different setup - constant viscosity, one fluid, thermal convection only. Thus we consider that the independence of melting efficiency on the mesh cell size is more important for this study.

l. 190 Please give a quantified range of temperature gradient here for consistency (as stated currently, the range is  $\geq$  11–33 °C/m, which is compatible with the rhyolite sill gradient). These lines l.190-192 stipulate after 35 years, that the temperature gradient is very close to 15°/m along a specific depth range.

1.212 °C/m/m  $\rightarrow$  °C/m. Corrected.

l. 214 (and also l. 290) It is important to state the reason(s) behind the crash: vanishing time step, stalling of the residuals, temperature runaway, instability of Eq (4), ... Currently, I can only infer that the crash probably does not occur because of too high a viscosity (see next comment).

Simulations with high Ra and high Pr number thermochemical problems remains challenging. Not all our simulations reached the final time. The time-steps diminishes during the first ca. 10 years of the

modelled run-time, and in successful cases it manages to rise back up again once the convective layers are stably growing. Several cases end up with extremely small the time-step, others in drastic oscillation in the time-step that finally leads to unphysical results. Bounding the time-step upper limit promotes better stability. Higher mesh resolution tends to produce more stable runs despite longer computational duration, indicating that small time and length-scale perturbations are properly resolved. At later stages when basalt becomes viscous and the viscosity contrast between the magmas becomes small the mixing intensity increases and, of course, this process is not captured by the VOF method. We insert these explanations, 1.342.

- l. 229 The maximum viscosity is said to have a minor effect on the results. Does it mean that it also does not affect crash time (see comment above)? Not really indeed, we have tried that without significant success.
- l. 265. "although the 1D model shows temperature gradients within convective regions". This is a major drawback as the variable of interest is the temperature gradient. Just looking at Fig. 9, I guess that the 1D gradients across the melt zone thickness are basically meaningless because they are so far away from the 2D gradients. If my guess is correct, I suggest adding a few sentences about this issue as it shows clearly the added value of this 2D study.

We agree that our simplified 1D model does not capture all the complexity of the 2D simulations. The important message is that here we directly parameterize the effective thermal conductivity and obtain high values of the Nu number. The presence of significant temperature gradients in melt zones means that 1D effective conductivity (of the order of 200 W/m/K) is smaller than the effective conductivity in 2D, and that the 2D model resolves the heat transfer on the required time and length scales. Temperature gradients in the unmolten part above the rhyolite layer and its thickness are still, well captured by the 1D model.

- l. 283. I did not understand the end of the sentence: by a comparable amount to what? The other reviewer also pointed this. We deleted the end of sentence, please refer to our previous answer, page 2.
- l. 285. I appreciate the discussion and evaluation of the effect of the basalt sill thickness. I wonder why the rhyolite sill thickness is not mentioned, and thus I suggest adding a few sentences about it. Borisova et al. (2023) tried with a sill thicker that the 300 m used here and found larger temperature gradients after the dreaded 35 yrs. I am not a fan of highlighting every parameter and asking to explore it further, but here the conclusion of the paper hinges on that single sill thickness. Presumably a sill of 350 m would bump the gradient from 15 to 16 °C/m, suddenly making Scenario 1 unquestionably valid?

That is a good point, thank you for raising it. We have been more careful now throughout the text, to clearly express those gradients in both cases now. Indeed, a thicker rhyolite sill may produce a reasonable temperature jump, but it could not be much thicker or else it would have been detected by geophysical methods. This is also now mentioned, l. 304 (discussion), l. 392 (conclusion).

l. 317 "best matches" is a simplistic assessment of the results. I suggest rephrasing that first § in terms of degrees of freedom and parameter ranges. As an example, here is what I have in mind: An extrapolation of our results suggests that Scenario 1 likely fits the observed values within a very narrow range of parameters, whereas our results for Scenario 2 covers a wider range of parameters that yield gradient comparable to those observed. [...] For all these reasons we prefer Scenario 2 over Scenario 1.

We agree that this is a better way to interpret our results and develop our argumentation. We rephrase the text accordingly in the discussion (1.305), and more specifically 1. 392 of the conclusion.

1. 330-344. The petrology § are not linked to the framework of this study. Please ensure that it is the case. For instance, fractional crystallization is incompatible with the model assumptions, which needs to be mentioned. Another example is that hydrothermal fluids were needed to produce the partial rhyolitic melts, but 1) the model ignores hydrothermal fluids and 2) the whole § on 1. 301-309 is dedicated to show that hydrothermal fluids played no direct role in [...] partial melting and the following reaction of the rock with [putative] basaltic magma. We rephrased, shortened and reorganised this section to make it easier to read. "Fractional crystallisation" has been removed. We try now to better link the justification that the numerical models ignore hydrothermal fluids interaction with melts, consistently with the petrological information that indicates that there hasn't been any.

Finally, the duration of 33 yrs is chosen here (and also in the Conclusions l. 354), whereas the whole work (starting l. 60) and all model results were evaluated at 35 yrs. This would be a detail if I were not tempted to wonder how much higher the 15°C/m gradient of Scenario 1 is at 95% of the simulation time. We rephrased the sentences where "33 yrs" appeared to make a more consistent link between modeling predictions and the time-lapse from the Krafla-Fires to the 2009 drilling date. On the other hand the uncertainties provided

by the modeling results (related to viscosities and mesh resolution) do not really allow us to discuss the variability between periods of 5% of time. Nevertheless we have smoothen our other statements favoring the basalt vs. rhyolite scenarios (cf. points and answers above).

Appendices. I appreciated the clever selection parameter sweeps that gave me confidence in the numerical outputs. Thank you!

l. 460. I was trying to find the answer to the question: how much, if any, total/dissolved water were assumed to be present in the rhyolite/basalt. Borisova et a., (2023) reports rhyolite composition with 1.9 wt% H2O, and, unless I got lost, no basaltic composition. This is a confusing issue as this water-bearing rhyolite is the result of partial melting but the injected sill is a different (source) rhyolite in Scenario 1. To clarify this issue, could you simply add a Supplementary/Appendix Table with the initial MELTS composition for both Scenarios? Yes compositions are either the felsic rock (also called granophyre when cold, rhyolite domain throughout the text either cold or partially molten), or the hot basaltic intrusion.

We include now in Appendix A a Table with the compositions extracted from MELTS that were assumed in the numerical models, which is part of a ms. that is in revision (Borisova et al., 2025, submitted to JVGR).

In addition to the modifications required by the reviewers above, we made two additional changes:

- we have suppressed column 2 in Table 3 that was showing local run names which helped us orient ourselves within our local directories. They are useless now in the ms.
- we have been able to run additional test cases with a basalt intrusion for longer time during the 3 months of the reviewing period; first, two low to moderate resolution cases could run up to 50 yrs, and the corresponding rhyolite melting zone thicknesses (MZT) plots are displayed now in Fig.6b. Finally an even higher resolution test at cell size 0.125 m could be ran until 20 yrs, which we now display in Appendix B, new figures B3 and B4. We also updated Figures B1 and B5, and 6b. These additional results strengthen our estimates of the temporal evolution of the MZT.

---

## Author Response (AR2)

Dear Editor,

We have done our best to correct the manuscript according to reviewers suggestions. In the uploaded version of the paper we have corrected the sentence according to your suggestion and added the reference to the relevant paper.

We hope to see our work published soon, All the best Muriel, Oleg and Anastassya